# Health Literacy and Critical Lecture as Key Elements to Detect and Reply to Nutrition Misinformation on Social Media: Analysis between Spanish Healthcare Professionals

**DOI:** 10.3390/ijerph20010023

**Published:** 2022-12-20

**Authors:** Sergio Segado-Fernández, María del Carmen Lozano-Estevan, Beatriz Jiménez-Gómez, Carlos Ruiz-Núñez, Pedro Jesús Jiménez Hidalgo, Invención Fernández-Quijano, Liliana González-Rodríguez, Azucena Santillán-García, Ivan Herrera-Peco

**Affiliations:** 1Department of Health Sciences, Universidad Europea de Canarias, Calle Inocencio García, 1, La Orotava, 38300 Santa Cruz de Tenerife, Spain; 2VALORNUT-UCM (920030) Research Group, Department of Nutrition and Food Science, Faculty of Pharmacy, Complutense University of Madrid, 28040 Madrid, Spain; 3Nursing Department, Faculty of Medicine, Alfonso X el Sabio University, Avenida Universidad, 1, Villanueva de la Cañada, 28691 Madrid, Spain; 4Phd Student Program in Biomedicine, Translational Research and New Health Technologies, School of Medicine, University of Malaga Blvr. Louis Pasteur, 29010 Málaga, Spain; 5Traumatology and Orthopedic Surgery Service, Hospital Universitario Nuestra Señora de Candelaria, Ctra. Gral. del Rosario, 145, 38010 Santa Cruz de Tenerife, Spain; 6Faculty of Psychology, Universidad Alfonso X el Sabio, Avenida Universidad, 1, Villanueva de la Cañada, 28691 Madrid, Spain; 7Nursing Department, Valencia International University, C/Pintor Sorolla 21, 46002 Valencia, Spain; 8Faculty of Health Sciences, Universidad Alfonso X el Sabio, Avenida Universidad, 1, Villanueva de la Cañada, 28691 Madrid, Spain

**Keywords:** detection, health misinformation, healthcare professionals, public health, social media

## Abstract

Health misinformation about nutrition and other health aspects on social media is a current public health concern. Healthcare professionals play an essential role in efforts to detect and correct it. The present study focuses on analyzing the use of competencies associated with training in methodology, health literacy, and critical lecture in order to detect sources of health misinformation that use scientific articles to support their false information. A qualitative study was conducted between 15 and 30 January 2022, wherein the participants were recruited from active users from a nutrition conversation on Twitter, diets, and cancer and defined themselves as healthcare professionals. This study demonstrates that health literacy and critical lecture competencies allow for the detection of more misinformation messages and are associated with a high rate of responses to users that spread the misinformation messages. Finally, this study proposes the necessity of developing actions to improve health literacy and critical lecture competencies between healthcare professionals. However, in order to achieve this, health authorities must develop strategies to psychologically support those healthcare professionals faced with bullying as a result of their activity on social media debunking health hoaxes.

## 1. Introduction

In recent times, with events such as the COVID-19 pandemic [1,2,3] which greatly impact the health of the global population, a phenomenon has been described in the diffusion of information. This phenomenon—already observed in previous catastrophes and epidemics—occurs when there is an increased need for information, such as the need to find health information [4,5,6]. However, in addition to this need, there is a vast quantity of information about health [7] that could be associated with the generation of confusion among the population and decreasing adherence to the recommendations of health authorities [8,9].

With respect to health information, one of the fastest ways to obtain information is through the internet, social networks, and cross-platform messaging apps such as WhatsApp or Telegram [10,11]. However, in addition to this immediacy, in these media formats, there is no control or verifiability of the contents; thus, they can become a rapid means of dispersion for unverified health information [12], generating misinformation around health [13,14] through the offer of content on social media that presents fictitious or incomplete arguments manipulating verified health content or fabricating data [15,16]. The rapid dissemination of this type of biased or misleading news throughout the digital world via social networks [13] significantly affects proper public health communication and diminishes self-care and individual health prevention measures [17].

Social media is where the population looked most frequently for health information during the COVID-19 pandemic, with a special emphasis on the link between COVID-19 and nutrition [18]. Given that, there is an increased need for information on pathologies such as obesity, diabetes, or cardiovascular diseases with a clear relationship with nutrition, in addition to aspects such as the possible role of food supplements in the activity of the immune system [18,19].

Nutrition is one of the topics that have seen an increase in misinformation messages and content, as has been observed in cases such as food supplements whose misuse can cause liver damage, or interactions with drugs such as anticoagulants in the case of vitamin K [18,20,21]. There is a close relationship between social networks and people’s food; as a way to obtain information that impacts people’s lives, it is of particular interest that they define elements that can recognize and filter unverified health information to help control the spread of misinformation [7].

Currently, the presence of different healthcare professionals on social networks is increasing [22,23,24,25] and, although this presence coincides in most cases with personal use [24,26] or its use as a profession-promoting channel during the COVID-19 pandemic [26,27], it is also used as a communication channel with patients [25] or the general population. In addition to the above, health professionals are often considered as opinion leaders or influencers on health issues [28], which can be attributed to the fact that they are considered to have good health literacy skills [28,29]. Thus, they can support their statements with highly reliable sources of information [23] and communicate them in a way that is adequate for the understanding of the population [30,31].

Due to the above, and given the role that health professionals have traditionally played in everything related to supporting individuals and communities in understanding messages related to their health care [32] and focusing on the public health of the population, health professionals may become essential in mitigating the diffusion of health misinformation [33,34], thus reducing the belief in unverified health information since health information disseminated by health professionals is considered highly reliable [35]. This coincides with what has been stated by numerous studies which stress the need for health professionals to act on the dissemination of misinformation either by stressing the need to have a greater presence on social networks [26] or by using it as a place to provide health education [36]. However, there are few studies which address how to manage the fight against misinformation in health; one of the first studies in this regard was carried out by Bautista et al. [16], where it was indicated that health professionals, physicians, and nurses dedicated time and effort to respond to the misinformation they found associated with their concern for public health.

In this sense, it should be noted that, although health literacy competencies are high, it is necessary to emphasize that health professionals themselves can be vulnerable to false information [37], and training in research methodology and communication of this collective is the best tool to differentiate between fake and verified information and to avoid the dissemination of inaccurate information [38]. Regarding misinformation on health, and nutrition specifically, it is important to highlight that scientific articles are often leveraged with the aim of giving the news a “cover” of scientific veracity, making it essential to always check the source and ensure it has a reliable basis [17]. In this situation, the possibility that the news items have links to scientific articles must be considered.

In the present study, the main objective was to analyze the relationship between knowledge of research methodology and the application of scientific evidence search to check the reliability of health information linked to scientific articles in nutrition. Furthermore, we also wanted to study how health professionals express their responses to the misinformation analyzed.

## 2. Materials and Methods

### 2.1. Procedure and Participants

We used a qualitative research approach to obtain the opinions of health professionals on how they manage health information on social networks, how they check the reliability of messages providing health information, and how they respond to health misinformation.

For the selection of the sample, the social network Twitter was used to find active users who sent messages about health or nutrition information. Active users were identified as those who generated tweets, retweets, etc., in a specific conversation, in Spanish, about nutrition, diets, and cancer. Users who identified themselves as health professionals and who had participated in that conversation were located. Information from tweets and user descriptions were extracted through an API (application programming interface) search tool, using the professional version of NodeXL software (Social Media Research Foundation, Redwood City, CA, USA).

Participants were recruited through Twitter direct messages, using a convenience sampling method. We found a total of 157 users contacted through the Twitter direct message tool, of which only 81 health professionals responded to our questionnaire, representing 51.59% of the total. It was found that 23 of the 81 participants were male (28.4%), while 58 were defined as female (71.6%). Regarding the age of the participants, it was found that most of them were between 36 and 50 years old (n = 38; 46.9%), followed by the participants under 35 years old (n = 35; 30.9%) (Table 1). When assessing the participants’ presence on social networks, they indicated that, in addition to being present on the social network Twitter, they were also present on the following networks: Facebook (n = 56;69.13%), Instagram (n = 60; 74.07%), TikTok (n = 15;18.52%), and LinkedIn (n = 28;34.56%).

Regarding the number of followers of the participants, the number of Twitter followers was assessed, finding that 51.85% (n = 42) participants were considered micro influencers (1000–5000 followers), followed by users with less than 1000 followers (n = 31; 38.27%), while influencers with more than 5000 followers represented 17.28% (n = 14) of the participants.

Regarding the use of social networks, the participants reflected that 72% (n = 59) used the networks for personal and professional purposes, while 18.5% (n = 15) reported strictly personal use and 8.6% (n = 7) used it in a strictly professional capacity.

### 2.2. Data Collection

The data collection was developed from 15 to 30 January 2022. To obtain these data, a survey was developed on two different sections, the first was designed to collect sociodemographic information from the participants focused on: (i) gender, (ii) age, (iii) social media usage, and (iv) followers on Twitter.

The second section included open-ended questions focused on the type of strategies they used to recognize the veracity of health information that reached them through social networks, as well as the way in which they managed this information.

The questions asked focused on five main areas of interest, developed from and based on Bautista’s previous studies [16]. The areas of interest were: (i) the participant’s presence and type of activity on social networks, (ii) strategies for recognizing the origin of health information, (iii) how health professionals handle the health information that reaches them through social networks, (iv) which actions they take when responding to information they consider incorrect, and (v) the importance of knowledge of the research methodology for filtering health information.

### 2.3. Procedure and Ethical Considerations

Participants were selected using a convenience sample, the participation was voluntary and, prior to the start of the responses, potential participants were provided with all the information related to the study. Participants gave their consent by checking a box designed for this purpose, which provided access to the survey.

The present study was previously evaluated and approved by the Research Ethics Committee of the Universidad Alfonso X el Sabio (reference 2022_2/129).

### 2.4. Data Analysis

The analysis of the data compiled was performed in several steps, and a content analysis was performed with the categories created after analyzing the data. A qualitative analysis was developed in the present study, and the codification used in the present study was based on a previous study [16] and those derived from the data collected considering the participants’ perspectives, experiences, and opinions. The main categories and subcategories are shown in Table 1.

It is important to note that in this study, content and category coding were performed independently by three researchers and corroborated by a fourth person, whereby any differences in approach and focus were always discussed and resolved with full agreement.

Finally, for data analysis, descriptive and inferential statistics were used via the Statistical Package for the Social Sciences software (SPSS) version 23.0 (IBM, Armonk, NY, USA). The categorical variables, included in the present study and derived from the subcategories defined in the qualitative approach, have been expressed as the total number of individuals and proportions. Furthermore, the comparison between groups was performed with a non-parametric test, the chi-square test, since this test does not require homoscedasticity in the data and permits the evaluation of dichotomous independent variables. The statistical level of significance was set at *p* < 0.05.

## 3. Results

### 3.1. Health Information on Social Media

The participants indicated that the process they follow when managing the health information that reaches them through social networks is as follows: First, they verify the origin of the information. Subsequently, they consider whether the sender of the information is reliable. After this, they will verify the health information contained in those messages, and then proceed to respond if the information is false or disseminate it if they consider it to be correct.

The method employed by the users to determine the reliability of the information is focused on the analysis of the external reliability of the health message received. This validation is based on (i) the subject matter of the message, (ii) its scientific and clinical coherence, (iii) the validity and reliability of the author, and iv) the existence of external links (it being important that these are not broken and lead to addresses other than those mentioned in the message). The results showed that 96.29% (n = 78) of participants who reviewed the health information that reached them through social networks indicated that they carried out an external review of the content.

In relation to the external analysis of the validity of the messages, it is worth noting that 28.2% of the participants (n = 22), indicated that they check the validity and reliability of the user issuing the research only in the case that it is not someone they know checks the health information that comes from. When the possible effect of the variables of age, sex, number of followers, or type of use of social media was assessed, it was found that a significant relationship was only observed between checking the information by origin associated with (i) age (χ^2^ = 13.867; *p* = 0.008) and (ii) social media use by participants (χ^2^ = 11.518; *p* = 0.021).

### 3.2. Analysis of Critical Lecture of Health Information Linked to Scientific/Technical Documents

First, the participants were asked if they had training health literacy, and 50 (61.72%) of them indicated that they had this type of training. We observed that the health literacy competence is not associated with any sociodemographic characteristic (Table 2).

This was divided into a review of the authorship of the message, a situation in which 71.79% (n = 56) indicated that they always carried out the review compared to 28.2% (n = 22) who indicated that they only reviewed the reliability of the author if they did not know him/her (Table 2).

Regarding the external verification of the message, the participants were asked about the review of scientific articles and the application of critical lecture. It was observed that 33.33% (n = 26) of the participants indicated that they were looking for the article but that they only reviewed it based on their clinical experience, not applying critical lecture analyzing the methodological characteristics of the article. In total, 66.66% (n = 52) indicated a critical reading of the articles or technical documents that were linked to these messages (Table 2).

In relation to the knowledge of methodology, it waas observed that it is not related to the (self-perceived) competence of health literacy (χ^2^ = 1.871; *p* = 0.384). However, it was found that the possession of this training was associated with the review of aspects of methodology such as study design and type of sample, as well as the statistical analysis used (χ^2^ = 10.15; *p* = 0.022).

### 3.3. Reply to Health Misinformation

Regarding the response to the messages that the participants consider as having disinforming health content, it was observed that 56.79% (n = 46) reply to the detected messages.

Among the participants who answered, 56.52% (n = 26) indicated that they wrote the messages in non-scientific language. The tone of the response was defined by 54.34% (n = 25) of participants as neutral, followed by conciliatory 36.95% (n = 17), and only 8.71% (n = 4) reported an aggressive or critical tone.

The rebuttal of messages with misinformation was mainly conducted in public—84.78% (n = 39)—to the user that sent the message. Finally, when participants were asked whether they include a scientific reference in the replies to misinformation messages, 71.74% (n = 33) of participants reported to include scientific references (Table 3).

The review of the results to assess the possible relationship of sociodemographic variables with responsiveness and how the messages are structured did not show a statistically significant relationship with the intention to respond to messages detected as health misinformation, the tone of the response message, the language used in the replies, the method employed by the participants to rebut the misinformation, or the inclusion of appropriate scientific references in the replies (Table 3). The exception is age, which shows a statistically significant association with the incorporation of references in the replies (χ^2^ = 9.763; *p* = 0.045) (Table 3).

### 3.4. Relationship of Critical Reading Skills with the Response to Health Misinformation

It was observed that the frequency and interest in verifying the information that arrives in health were not related to the knowledge of the research methodology indicated by the participants. However, it was significantly associated with the application of the participant’s critical reading skills (c = 0.254; *p* = 0.018).

Likewise, it was observed that the development of critical reading of scientific articles linked to health messages was significantly associated with the response to these messages, as well as with the tone of the responses, the language used, and the inclusion of scientific references in the responses.

The intention to respond to misinformation was statistically significantly associated with the incorporation of scientific references, and the use of a neutral tone or non-technical language (Table 4). Finally, it can be found that the tone of the messages includes the language used (without technical language). The incorporation of scientific references in the responses to the messages with misinformation is not associated with the tone of the messages or with the type of language used (Table 4).

## 4. Discussion

The present study offers an insight into how Spanish healthcare professionals present on the social network Twitter react to messages with a misinformation approach to health.

Nowadays, health information reaches both users who are health professionals and the general population very quickly and easily [11]; especially serious is the arrival of messages that may include links to technical reports or even scientific articles that may have been altered or distorted to support the information sent in the message that will generate misinformation [15,16].

Although healthcare professionals are considered as professionals with high competencies in health literacy [29,34], it is their role on social networks where it can be essential in the face of the lack of control of the veracity of health content that occurs on social networks [12]. On numerous occasions, health professionals have been called for as the agents who can verify the veracity of health information on social networks [5,39,40,41], and thus help the population to maintain better information and facilitate making the best decisions about care [42]. This is of great importance in areas such as nutrition where it is found that social media can influence the approach to different pathologies, such as nutritional disorders, either aggravating or even triggering the pathology due to the misinformation received [43,44].

In the present study, we found that a high number of health professionals act in this manner, reviewing and replying to the uninformed messages that reach them, which coincides with the findings of other authors [16]. Although it is common for social networks to be used from a personal point of view [24,25] by users, health professionals are characterized by a high level of commitment to the health of the population [16,26,27].

As has been observed, the knowledge of having received training in health literacy and having critical reading skills makes it easier for health professionals to respond more assiduously to messages that can be considered as misinformation. This may be associated with a higher degree of self-efficacy and confidence [45] in detecting the scientific reasons why information is untrue [39]. This ability and the ability to respond to misinformation were not found to be related to the sex of the participants in the present study, despite the fact that the health professions are largely occupied by women [46]. It is of particular importance to note that the inclusion of scientific references in the answers given when detecting health misinformation messages is associated with age, more specifically the group between 36 and 50 years of age. This situation is consistent with the majority group of Twitter users in Spain, who are between 32 and 46 years of age, with 39% of users [47]. Likewise, and associated with age, it was also observed that there was a greater ability to develop searches and checks of the origin of information, which is consistent with the greater likelihood of having received training in these areas.

However, it seems essential to address the handicap for the review and non-dissemination of health misinformation on social media that it is necessary to always check messages regardless of their origin. Although some studies indicate that health professionals always check the information that reaches them [16], in the present study, it was observed that a high percentage of health professionals did not check the information that reached them from known users. This situation contrasts with the need to have a critical attitude that makes it possible to detect and correct [16,48], as soon as possible, false information that could spread [12,13] and generate health problems for the population [7,18].

In relation to the response to health misinformation messages, although this corrective activity is associated with professional identity, as a healthcare professional [49], it is important that professionals have adequate communication skills both at a personal level and for social media [50]. It should also be considered that detecting false information and deciding not to disseminate it should not be treated in the same way as actively acting and publicly replying to false content.

The step of replying exposes the professional to users who may attack or harass him or her on social networks [16,51,52]. The possibility of suffering this type of harassment can affect clinical practice [51] and even the personal life of health professionals due to the stress and anxiety caused [5,33]. It is important to note that the public health service provided by health professionals on social networks is carried out on a completely voluntary basis [51].

Finally, it should be noted that this study has several limitations, mainly related to the study design. Furthermore, the participant selection criteria, such as having only used Twitter, limited the potential participants in the present study. Furthermore, retrieving information using a specific hashtag and keyword may have missed users who posted messages without using this keyword. Thirdly, the study sample, due to the nonprobability sampling used, was not representative. Therefore, future research should improve the sampling technique to avoid possible biases.

Furthermore, this study has several strengths. Firstly, this study showed the strategies used by Spanish healthcare professionals to recognize and manage health disinformation received through social media. Secondly, we observed the main communication strategies used to reply to health disinformation, and, thirdly, we explored the opinion of health professionals about the importance of knowledge and management of research methodologies for the management of health misinformation.

## 5. Conclusions

We consider that social media plays an important role in our society and currently represents one of the main sources of information and dissemination regarding how to follow healthy eating and living habits. Thus, misinformation about nutrition on social media is a problem that affects a large part of the population and can modify the nutritional patterns of the population. This situation could affect people’s health, and thus represents a public health problem.

In this study, we reveal two of the main competencies of those who actively participate on social networks in terms of detecting and responding to disinformation messages. Together, health literacy and critical lecture are the main tools that can help health professionals to achieve the confidence to respond to these messages.

However, these being two essential elements in the fight against health misinformation on social networks, it is important to point out that health professionals act on a voluntary basis.

Regarding encouraging the participation of healthcare professionals in the fight against health disinformation on social networks, there is a very interesting debate. At least to the authors’ knowledge, there are no institutional training and support initiatives for health professionals focused on the active fight against health misinformation; many of them carry out this work altruistically, with the few means at their disposal.

We believe that part of the improvement of the fight against misinformation involves improving the training of healthcare professionals in critical scientific lecture skills.

To enhance this activity, and thus increase the effectiveness of the fight against misinformation, we propose that healthcare institutions use three strategies: (i) increase the training of practicing healthcare professionals and students in health literacy and critical reading; (ii) develop programs that support healthcare professionals active on social networks from a psychological point of view, due to the pressure they may receive for their altruistic activity for the good of the population; (iii) receive adequate training on the proper way to communicate, particularly the essential incorporation of communication professionals who train health professionals in the best way to define the messages that dismantle incorrect or fraudulent health information.

## Figures and Tables

**Table 1 ijerph-20-00023-t001:** Main categories and subcategories defined in the present study.

Categories	Subcategories	
Filter strategies of health misinformation	Origin of health information	Authentication of users
	Check the internal reliability	
	Elements to check the external reliability	Tone of messages Presence or absence of data to support the affirmation about health. Presence of links to scientific or technical documents
Check the reliability of scientific papers linked to health information messages	Health literacy competence	Auto perception of high or low competence on health literacy
	Critical lecture competence	Analyzed the methodology and data analysis used Check reliability using, exclusively, clinical experience (no application of methodology research or data analysis competencies)
Reply to health misinformation messages	Frequency of reply	
	Tone and (kind of) language used	Technical/non-technical
	Rebuttal misinformation	Publicly or privately
	Inclusion of scientific references in the replies	

**Table 2 ijerph-20-00023-t002:** Influence of sociodemographic characteristics on the frequency of checking the reliability of health information with scientific papers linked.

		Competence on Health Literacy			Critical Lecture of Articles Linked to Health Messages		
		Yes	No	(χ^2^; *p*-value)	R.M.	C.E.	(χ^2^; *p*-value)
Gender	Female	33	25	(2.019;0.121)	37	21	(0.0154; 0.558)
	Male	17	6		15	8	
Age (years)	<35	12	13	(2.901; 0.234)	14	11	(3.821; 0.148)
	36–50	26	12		23	15	
	>50	12	6		15	3	
Social media use	Personal	9	6	(0.31; 0.856)	9	5	(0.275; 0.871)
	Professional	5	2		5	2	
	Mixed	36	23		38	21	
Followers	<1000	23	12	(0.424; 0.809)	22	13	(0.200; 0.905)
	1000–5000	21	15		24	12	
	>5000	6	4		6	4	

Where: (i) R.M. means that healthcare professionals applied critical lecture to the scientific paper focusing on methodology and analysis aspects. (ii) C.E. means healthcare professionals search the scientific article but check the reliability using only their clinical experience to check the reliability of the health information.

**Table 3 ijerph-20-00023-t003:** Influence of sociodemographic characteristics on how users reply to health misinformation messages.

		Reply to Health Misinformation Messages			Tone of the Replies				Language Used in the Replies			Rebuttal of Misinformation			Included Appropriate Scientific References in the Replies		
		Yes	No	(χ^2^)	Conciliatory	Critical	Neutral	(χ^2^; *p*)	Scientific and Technical	Non-Scientific	(χ^2^; *p*)	Publicly	Privately.	(χ^2^; *p*)	Yes	No	(χ^2^; *p*)
Gender (n; %)	Female	33	25	(0.001; 0.584)	13	2	17	(1.817; 0.611)	15	19	(0.054; 0.974)	28	5	(0.013; 0.994)	22	10	(0.230; 0.891)
	Male	13	10		4	2	8		6	7		11	2		11	3	
Age (years)(n; %)	<35	14	11	(0.036; 0.982)	5	4	5	(11.978; 0.062)	7	7	(5.472; 0.242)	15	2	(0.450; 0.978)	6	8	(9.763; 0.045 *)
	36–50	22	16		9	0	13		6	16		18	4		18	3	
	>50	10	8		2	0	8		8	3		10	1		8	3	
Social media use	Personal	9	6	(7.228; 0.300)	4	1	3	(7.228; 0.300)	5	2	(0.936; 0.919)	9	0	(3.667; 0.453)	5	4	(8.480; 0.075)
	Professional	4	3		2	2	1		2	4		4	0		1	3	
	Mixed	33	26		11	1	21		13	20		25	8		26	7	
Followers	<1000	20	15	(5.943; 0.430)	7	0	13	(5.943; 0.430)	9	11	(0.269; 0.992)	17	3	(0.238; 0.993)	13	6	(2.077; 0.722)
	1000–5000	21	15		8	4	9		9	12		18	3		15	7	
	>5000	5	5		2	0	3		2	3		4	1		5	0	

* means *p* < 0.05.

**Table 4 ijerph-20-00023-t004:** Association between competencies associated with research methodology and the reply to misinformation.

	A	B	C	D	E	F	G
A							
B	0.11; (n.s.)						
C	0.25; *	0.53; ***					
D	0.21; (n.s.)	0.38; ***	0.53; ***				
E	0.21; (n.s.)	0.41; **	0.53; ***	0.71; ***			
F	0.22; (n.s.)	0.42; ***	0.55; ***	0.70; ***	0.74; ***		
G	0.21; (n.s.)	0.43; ***	0.55; ***	0.70; ***	0.13; (n.s.)	0.25; (n.s.)	

Where: A, checks the reliability of information; B, competence on health literacy; C, competence on critical lecture of scientific articles/technical documents; D, replies to misinformation detected; E, tone of replies; F, language used in replies; and G, the inclusion of scientific references in replies. n.s., means non-significant; * means *p* < 0.05; ** means *p* < 0.01; *** means *p* < 0.001.

## Data Availability

The data that support the findings of this study are available from the corresponding author upon reasonable request.

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
