# Peer review of "Health Literacy and Critical Lecture as Key Elements to Detect and Reply to Nutrition Misinformation on Social Media: Analysis between Spanish Healthcare Professionals"

_ijerph, 2022, doi:10.3390/ijerph20010023_

Round 1

Reviewer 1 Report

Social media is used popularly in the world. It has brought great convenience for people to share information and obtain information. However, due to the lack of effective supervision mechanism, misinformation is widely spread on the network. That cases infodemic. Infodemic brings difficulties for people to find reliable information effectively. Sometimes misinformation, especially heath misinformation could possibly harm people’s heath. This paper investigates how nutrition professionals verify health information and how to response to the misinformation. The research question is important and necessary. However, the paper needs to be revised to reach the standard for publishing.

 1. The research recruited participants from Twitter. How to judge the active users mentioned in the research needs to be explained. It didn’t define the active users.

2. The study used convenient sampling, with 81 subjects. The sample size is small. The findings are not convincing.

3. The study provides open-ended questions to collect data from participants. It’s not clear that questionnaires or interview guidelines are distributed to the participants. Is it a quantitative study or qualitative study? It uses content analysis. However, the paper doesn’t give details about the process of content analysis. I can’t find what the content categories that authors have developed. It’s unclear how many categories was coded.

4. In terms of data analysis, further and more in-depth analysis can be conducted. For example, 3.2 only analyzed whether the origin of information would be verified. Criteria for judging is not discussed, such as the source of information, whether the information provided references, or the logic relationship between arguments etc. The paper only mentions that professionals only verify information when they do not know the author. The findings are relatively simple.

5. The participants are all Twitter users. I suggest the authors to collect data of these participants from Twitter to see how they response to the misinformation. The results of user self-evaluation may differ from the actual situation.

6. Motivation is important to trigger people’s behavior. What is the motivation for professionals to verify and correct misinformation? If this information is not relevant or important to individuals, will professionals keep verify information?

Overall, I think this study has selected an interesting topic, but the issues discussed are not to the point.

Author Response

  1. The researchers recruited participants from Twitter. How to judge the active users mentioned in the research needs to be explained. It didn’t define the active users.

Reply to the reviewer: Thank you very much for your comments, undoubtedly indicating what has been considered an "active user" is essential to understand that these users and why they are important to understand the objective of the study.

To solve the situation described by the reviewer, the manuscript has been rewritten, in the participants section, to explain more accurately what was considered an active user. 

As well as a better explanation of the Twitter conversation that allowed the identification of these health professionals. 

The information was included in a new section called “Procedure and participants”.

We have modified the abstract to include the information included in the manuscript. about the active users and how we identified them.

  1. The study used convenient sampling, with 81 subjects. The sample size is small. The findings are not convincing.

Reply to reviewer: Thank you very much for your indications. The fact that we use a non-probability sampling method means that our findings are not extrapolated to the population. We are aware of this reality, so to solve these situations we have re-write the manuscript to include this limitation, in a clear way, in the “limitations” paragraph included in the discussion section.

Furthermore, an explanation of the small sample size is provided in the “procedure and participants” section.

Finally, the abstract has been modified to reflect the changes in the information included in the manuscript. This will facilitate the understanding of the study for future readers.

  1. The study provides open-ended questions to collect data from participants. It’s not clear that questionnaires or interview guidelines are distributed to the participants. Is it a quantitative study or a qualitative study? It uses content analysis. However, the paper doesn’t give details about the process of content analysis. I can’t find the content categories that the authors have developed. It’s unclear how many categories were coded.

Reply to the reviewer: You are absolutely right in indicating that it has not been clearly defined what type of study has been developed.

To avoid this situation and to provide all the information necessary for future readers to understand what has been developed in this study, the procedure section has been rewritten to clearly state that it is a qualitative study.

Likewise, and given that the information related to the qualitative analysis developed was not clear, all the information related to this situation, as well as the categories and subcategories of interest for this study, has been specified in the "Data analysis" section.

  1. In terms of data analysis, further and more in-depth analysis can be conducted. For example, 3.2 only analyzed whether the origin of information would be verified. Criteria for judging are not discussed, such as the source of information, whether the information provided references or the logical relationship between arguments, etc. The paper only mentions that professionals only verify information when they do not know the author. The findings are relatively simple.

Reply to the reviewer:

Thank you very much for your comments. Undoubtedly, not having included the elements that make a health professional decide on the validity of a news item is an error that does not facilitate the understanding of the information to be included in the manuscript.

The structure of the results of the article has been modified so that, among other changes, information has been included on what elements the participating health professionals review when deciding on the validity of health information that reaches them through social networks. 

  1. The participants are all Twitter users. I suggest the authors collect data on these participants from Twitter to see how they respond to the misinformation. The results of user self-evaluation may differ from the actual situation.

Reply to the reviewer:

Undoubtedly, the collection of information, focusing on Twitter users, is one of the elements that have been defined in the limitations of the study. Modifying the discussion to include this limitation.

Likewise, the research design itself, where information is obtained through a self-perception of how the detection of misinformation is managed and the way in which it is responded to, is another limitation. It has also been included in the limitations section of the discussion.

The next project to be addressed is the analysis of the way in which these users, self-defined as health professionals, actually respond to health misinformation. By conducting an observational study of a specific conversation and assessing the interactions of health professionals.

  1. Motivation is important to trigger people’s behavior. What is the motivation for professionals to verify and correct misinformation? If this information is not relevant or important to individuals, will professionals keep verifying information?

Reply to reviewer:

Thank you for your contributions to improving the focus of the conclusions of the article.

This section has been re-structured to reflect more adequately the existing problem, as well as the lack of institutional support and the need to receive support and training in 3 main areas: theoretical and practical knowledge in research methodology, support in the structure, and preparation of messages to respond to misinformation. Without forgetting the psychological preparation.

Reviewer 2 Report

Thank you very much for the opportunity to review the study.

After reviewing the content and information provided, here are my comments:

1. the abstract should include more information about the methodology of the study, but no information about the group size.
2. The introduction is too long and the literature should be given at the end of the sentence/ paragraph.
3. numbering should not be used such as in the following excerpt
"From the above, in the present study the main objective were i) to analyze the reltionship between knowledge of research methodology and the application of scientific evidence search to check the reliability of health information linked to scientific articles in the area of nutrition, ii) and to study how health professionals express their responses to the misinformation analyzed "
4. no information in the text regarding Figure 1. what does it represent, etc.?
5. Table 1 is completely illegible, its structure should be improved.
6. Please describe in detail the use of statistical methods.
7. Table 2 duplicates the information described in the text.
8. Information about the study group should be in the methodology section.
9. Study group is small given the CAWI method used.
10. After the percentage, the number should be given as n=.
11. Table 3 is unreadable.
12. Table 4. remove p-value and use a footnote. Reduce the number of decimal places. Clarify "c".
13. Conclusions are not very readable, no specific conclusions that are revealing, contribute something significant.

Author Response

  1. the abstract should include more information about the methodology of the study, but no information about the group size.

Reply to reviewer: Thank you very much for your indications. We have rewritten the abstract to include the changes suggested.  

  1. The introduction is too long and the literature should be given at the end of the sentence/ paragraph.

Reply to reviewer: Thank you very much for your opinion, undoubtedly the translation was very long and could not offer the right information. We have rewritten the text to reduce it significantly.

The bibliography has been included at the end of each idea or piece of information provided in order to facilitate better understanding for future readers.

  1. numbering should not be used such as in the following excerpt
    "From the above, in the present study the main objective were i) to analyze the relationship between knowledge of research methodology and the application of scientific evidence search to check the reliability of health information linked to scientific articles in the area of nutrition, ii) and to study how health professionals express their responses to the misinformation analyzed "

Reply to reviewer: Thank you very much for your indications. 

In relation to the objectives, we made a mistake and don’t explain correctly that we have two main goals with this study. We have re-write the objectives to eliminate numbers.

  1. no information in the text regarding Figure 1. what does it represent, etc.?

Reply to reviewer: After reviewing again the structure of the material and methods section, we have observed that there is no sense in including Figure 1, which has been removed from this version of the manuscript.

  1. Table 1 is completely illegible, its structure should be improved.

Reply to reviewer: Table 1, from the previous version, has been removed from the new manuscript version.

A new table 1 has been included, that contains information about the categorization of the information provided by the participants. 

The purpose of this is to clarify the information provided to future readers, which was not clear in the original Table 1 and could generate confusion.

  1. Please describe in detail the use of statistical methods.

Reply to reviewer: Thank you very much for your contributions to improving the quality of the manuscript.

The part relating to the statistical analyses carried out in the study has been rewritten to indicate clearly and simply what has been done and how the categorical variables included in the study have been expressed.

  1. Table 2 duplicates the information described in the text.

Reply to reviewer: You are absolutely right with your point. To avoid this duplication we have deleted table 2 from the manuscript.

  1. Information about the study group should be in the methodology section.

Reply to the reviewer: It certainly makes more sense to include the sample information in the methodology section. 

For this purpose, we have removed the "Sample description" section from the Results and moved it to the Procedure and Participants section in the Material and Methods section.

  1. The study group is small given the CAWI method used.

Reply to reviewer:

This was an unforgivable error since a previous template with this information was used.

In the present study, potential participants were contacted via Twitter direct message and were provided with a link to provide the information, after accepting their participation with consent to participate.

To define this clearly, the Material and methods section has been completely rewritten, as it was found that this problem existed

  1. After the percentage, the number should be given as n=.

Reply to reviewer: Thank you very much for your indications. the manuscript has been revised and rewritten to include your suggestion.

  1. Table 3 is unreadable.

Reply to reviewer: Thank you very much for your contribution, since the table seemed to have a great complexity, the structure of the table has been modified to make it easier to understand. 

We have divided your information into two different tables and adjusted them to independent headings in the results section.

  1. Table 4. remove p-value and use a footnote. Reduce the number of decimal places. Clarify "c".

Reply to reviewer:

Table 4 has been rewritten to eliminate the p-value. Also, the meaning of "C" has been redefined for a better understanding by future readers. The number of decimal places has been reduced to 2.

  1. Conclusions are not very readable, no specific conclusions that are revealing, contribute something significant.

Reply to reviewer:

Thank you for your contributions to improve  the conclusions of the article.

This section has been re-structured to reflect more adequately the existing problem, as well as the lack of institutional support and the need to receive support and training in 3 main areas: theoretical and practical knowledge in research methodology, support in the structure, and preparation of messages to respond to misinformation. Without forgetting the psychological preparation.

We hope that this re-structuring of the conclusions will facilitate the reading and understanding of these conclusions by future readers. 

Round 2

Reviewer 2 Report

Dear Authors,

thank you very much for considering my comments. I believe that in its current version the manuscript meets the requirements and I support its publication.